# Inter-individual stereotypy of the *Platynereis* larval visual connectome

**Nadine Randel†, Réza Shahidi†, Csaba Verasztó, Luis A Bezares-Calderón, Steffen Schmidt, Gáspár Jékely\***

Max Planck Institute for Developmental Biology, Tübingen, Germany

**Abstract** Developmental programs have the fidelity to form neural circuits with the same structure and function among individuals of the same species. It is less well understood, however, to what extent entire neural circuits of different individuals are similar. Previously, we reported the neuronal connectome of the visual eye circuit from the head of a *Platynereis dumerilii* larva (*Randel et al., 2014*). We now report a full-body serial section transmission electron microscopy (ssTEM) dataset of another larva of the same age, for which we describe the connectome of the visual eyes and the larval eyespots. Anatomical comparisons and quantitative analyses of the two circuits reveal a high inter-individual stereotypy of the cell complement, neuronal projections, and synaptic connectivity, including the left-right asymmetry in the connectivity of some neurons. Our work shows the extent to which the eye circuitry in *Platynereis* larvae is hard-wired.

## Introduction

Innate stereotypical behaviors commonly observed in animals (*Jarrell et al., 2012*; *Kabra et al., 2013*; *Berman et al., 2014*; *Vogelstein et al., 2014*; *Zhang et al., 2014*) rely on the precise wiring of neuronal circuits during development. However, the stereotypy of neuronal circuits between individuals of the same species at the level of synaptic connectivity is not well known. Previously, several studies have addressed nervous system stereotypy at the level of single neurons, neuronal projection patterns, or neuronal activity. For example, in the zebrafish larval brain, neuronal spatiotemporal activity is highly stereotypical between individuals (*Portugues et al., 2014*). The fruit fly *Drosophila melanogaster* has stereotypical neuron types, axonal projection patterns, neuronal activity patterns and patterns of synaptic connectivity (*Yu et al., 2010*; *Mosca and Luo, 2014*; *Zhang et al., 2014*). A recent connectomic study found that the connectivity of interneurons mediating rolling behavior in *Drosophila* larvae were reproducible between two individuals (*Ohyama et al., 2015*). Nematodes also have a nervous system that is considered highly stereotypical (*White et al., 1986*).

Other studies that quantitatively addressed the stereotypy in connectivity patterns have found considerable variation among individuals. In the fly, inter-individual comparisons of local interneurons in the antennal lobe (*Chou et al., 2010*), as well as comparisons of segmentally repeated motor neurons (MNs) in the same individual (*Couton et al., 2015*) revealed considerable variation in finer-scale connectivity. Likewise, in mice, there is significant structural variation in connectivity patterns of MNs innervating the interscutularis muscles between the left and right sides of the same animal (*Lu et al., 2009*).

Here, we use serial-section transmission electron microscopy (ssTEM) to quantify the synaptic level inter-individual stereotypy of an entire sensory-motor circuit, the visual eye circuit of larvae of the marine annelid *Platynereis dumerilii* (*Randel et al., 2014*). Previously, we reported the neuronal connectome of this circuit, reconstructed from a 72 hr post fertilization (hpf) larva. *Platynereis* larvae develop following a strict cell lineage (*Fischer and Arendt, 2013*) and different individuals of the

**\*For correspondence:** gaspar. jekely@tuebingen.mpg.de

**†**These authors contributed equally to this work

**Competing interests:** The authors declare that no competing interests exist.

same developmental stage have a stereotypical complement of neurons (*Tomer et al., 2010*). However, whether synapse-level connectomes are reproducible among different *Platynereis* individuals is unclear. To test the stereotypy of the neuronal projection patterns and synaptic connectivity of individual neurons, we have reconstructed the visual eye circuitry of a second individual from the same batch, allowing the detailed comparison of two larval connectomes in *Platynereis*.

## Results

### Reconstruction of the visual eye circuit from a full-body ssTEM dataset of a *Platynereis* larva

We acquired a full-body ssTEM dataset of a 72 hr post fertilization (hpf) *Platynereis* larva (HT9-4) (*Figure 1A*), derived from the same batch as the previously described larva (HT9-3) (*Randel et al., 2014*). We imaged 5056 sections (40 nm/section) using conventional TEM. In this dataset, we traced the neuronal circuitry downstream of the visual eyes. The visual eye circuit of the second larva consists of 106 neurons, plus several ciliated and muscle effector cells (*Video 1*, *Source code 1*, and *Figure 1—source data 1*).

### Inter-individual stereotypy of the connectomes

The gross anatomy and cell complement of the HT9-4 eye circuit is very similar to that of HT9-3 (*Figure 1C*). We identified all cell types and groups of cells that we described for HT9-3. All cell types showed similar overall anatomical arrangement in the two individuals. For some cell types, including $IN^1$ interneurons and six ventral MNs ($MN^{l1-3}$ and $MN^{r1-3}$), we could correlate the two individuals at a single-cell level.

To test the stereotypy of synaptic connections, we compared patterns of connectivity between the two individuals (*Figure 1B,C*, *Figure 2B*, *Figure 1—figure supplement 1*, *Figure 2—source data 1*). We used several measures to compare the two connectomes. First, we defined a grouped connectivity matrix for HT9-4, similar to that of HT9-3 (*Randel et al., 2014*) (*Figure 2—source data 1*). We then correlated the two grouped matrices and found a strong correlation (Spearman's r = 0.62, p = 0.0001). For a more detailed comparison, we calculated the geometric mean of the two connectivity matrices and generated a combined matrix (*Figure 2A*). In this matrix, only connections are shown that are present in both animals. The combined matrix is similar to the individual matrices and we could find all major connections.

We also identified all reciprocal connections for HT9-4 and found the same strong reciprocal $IN^1$ motif as that of HT9-3 (*Figure 2—figure supplement 2*, *Figure 2—source data 1*). Additionally, we scored the correlation of the synaptic maturation of the HT9-4 photoreceptor cells (PRCs) with photoreceptor-cell rhabdom volume and axon length and found similar relationships as in HT9-3 (*Figure 3—figure supplement 1*, *Figure 3—figure supplement 1—source data 1*). All these measures indicate that the *Platynereis* eye circuit is strongly stereotypical at the global level of its synaptic connectivity.

Next, we focused on the stereotypy of single neuron types. First, we analyzed the correlation of connectivity between corresponding neuron types of HT9-3 and HT9-4.

We found significant correlations between both pre- and post-synaptic submatrices for most neuron types (*Figure 2C*). The neurons that did not show significant correlations between the two individuals were those with weak connections to the eye network, with only a few synapses in one or both individuals (e.g., $IN^{dcr}$ $IN^{vcl}$) or those that were truncated in HT9-3 ($MN^{r3}$, $MN^{l3}$).

To further test stereotypy at the single neuron level, we looked at the projection pattern and the distribution of pre- and post-synaptic sites in corresponding neurons between the two individuals. All neuron groups had similar projection patterns and spatial distribution of pre- and post-synaptic sites (*Figure 2—figure supplement 1*).

We also analyzed certain circuit elements in more detail. We first compared the connectivity of the visual eye PRCs (*Figure 3A*) and $IN^1$ cells. The visual eye PRCs showed very similar connectivity with the $IN^1$ interneurons in both animals. The strongest connection for all eyes was from PRCs to the cross-wise $IN^1$. However, we also identified a weak innervation of the ipsilateral $IN^1$ by the posterior eye PRCs that was consistent between the two animals (*Figure 3A*).

Next, we analyzed the pre- and post-synaptic partners of the MN cells. All MN cells, with the exception of $MN^{r3}$, are strongly innervated by 2–3 ipsilateral, plus 2–3 contralateral, $IN^{sn}$ cells in both

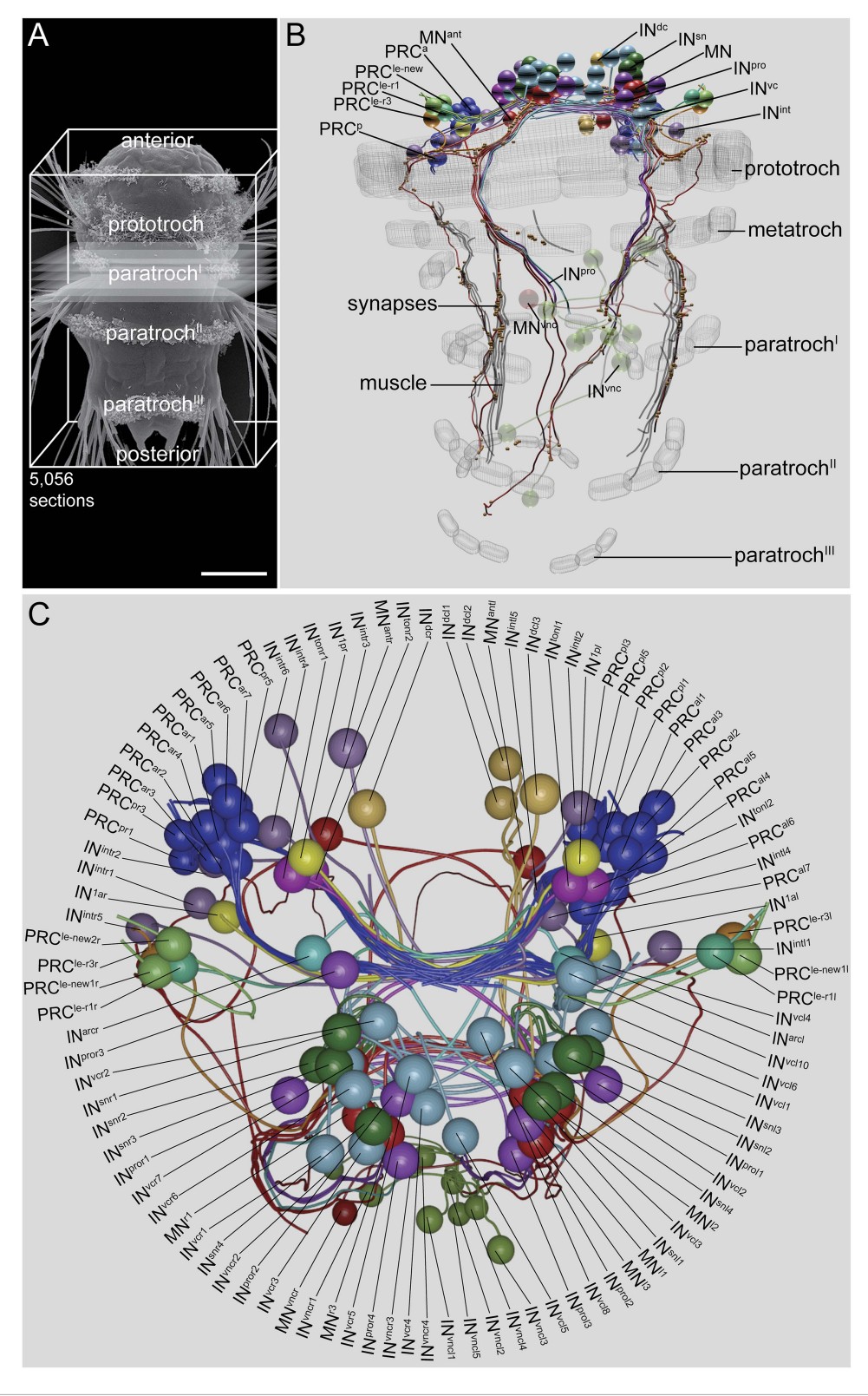

**Figure 1**. Visual eye and larval eyespot circuit reconstructed from a full-body dataset (HT9-4). (**A**) Scanning electron micrograph of a 72 hpf larva, dorsal view. (**B**, **C**) Blender visualization of the cell complement in ventral (**B**) and anterior (**C**) views. IN[1], primary interneuron; IN[arc], arc interneuron; IN[dc], dorsal interneuron; IN[int], intrinsic interneuron;
*Figure 1. continued on next page*

*Figure 1. Continued*

IN$^{sn}$, Schnörkel-interneuron; IN$^{ton}$, trans-optic neuropil interneuron; IN$^{pro}$, projection interneuron; IN$^{vc}$, ventral interneuron; IN$^{vnc}$, ventral nerve cord interneuron; MN, ventral motor neuron; MN$^{ant}$, anterior motor neuron; MN$^{vnc}$, ventral nerve cord motor neuron; PRC, photoreceptor cell of the visual eye; PRC$^{le-r1/3}$, photoreceptor cell of the larval eye, expressing *r-opsin3* or *r-opsin1*; a, anterior; p, posterior; l, left; r, right. Scale bar in (**A**) 50 μm.

The following source data and figure supplement are available for figure 1:

**Source data 1**. Layer accounts of the sections and images of HT9-4.

**Figure supplement 1**. Full connectivity matrix of the visual and larval eye circuit of the fully reconstructed larva HT9-4.

animals (*Figure 3B*). MN$^{r3}$ is also unique in another aspect of its connectivity. This neuron in both animals consistently displayed a highly left-right asymmetric connectivity pattern. MN$^{r3}$ is anatomically similar to MN$^{l1}$. Both cells are ciliomotor and muscle MNs, with strong connections to contralateral prototroch cells, to the longitudinal muscles and the paratrochs. However, MN$^{r3}$'s connectivity is unique in forming several synapses on contralateral MN cells (*Figure 3C*), being thus the only MN to potentially influence both contralateral and ipsilateral effectors. We could detect this asymmetric connectivity pattern for MN$^{r3}$ in both individuals. These results show that synaptic stereotypy can be found at the single neuron level and also applies to left-right asymmetric motifs in the network.

Our analysis shows that the visual eye connectome in *Platynereis* larvae has strong stereotypy both globally and at the single-neuron level. We could demonstrate this for patterns of neural projections, synapse distribution along neurites, identity of synaptic partners, strength of connections, and left-right asymmetries in connectivity. These results place the visual eye circuitry in a full-body context and establish the cellular-level stereotypy of neuronal circuitry in *Platynereis* larvae.

## Differences between the two datasets

The full-body dataset of HT9-4 allowed us to trace the projections of the ventral MNs in the trunk of the larva, a region that was not sectioned in HT9-3. This extended tracing revealed the presence of many more MN synapses to the trunk longitudinal muscles (maximum 68 synapses/MN) than were detected in HT9-3 (maximum 13 synapses/MN).

In HT9-4, we also identified a new interneuron type, which we named projection interneuron (IN$^{pro}$). Like the MNs, IN$^{pro}$ cells receive input from IN$^{sn}$ cells and project to the trunk at a more medial position. In the trunk, IN$^{pro}$ cells form weak connections to a group of ventral nerve cord interneurons (IN$^{vnc}$). The IN$^{pro}$ and IN$^{vnc}$ neurons are only weakly connected and likely represent a developing trunk circuitry. Based on new information from HT9-4, some of the cells previously described as MNs in HT9-3 have been reclassified as IN$^{pro}$. IN$^{pro}$ cells have somas near the MN somas and also project to the larval trunk, as revealed by the full-body dataset of HT9-4. In HT9-3, we classified 11 cells as MNs based on incoming synapses, soma positions, and projection patterns, but five of these cells did not form synapses on effector cells. Initially, we assumed these missing connections likely occurred in trunk sections posterior to the first segment in HT9-3. However, based on the data from HT9-4 (*Figure 1B*), these neurons more

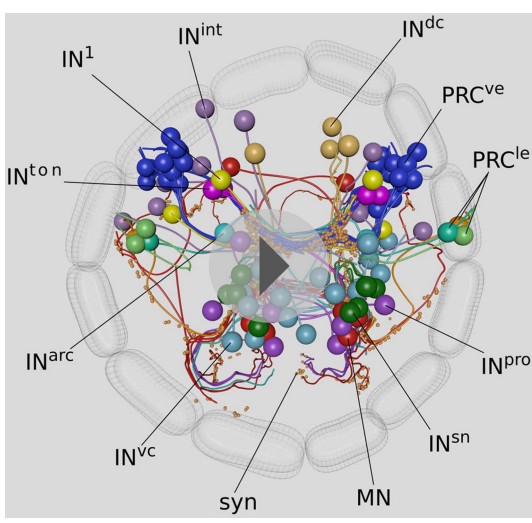

**Video 1.** 3D visualization of the cell complement of the *Platynereis* larval visual and larval eye circuit.

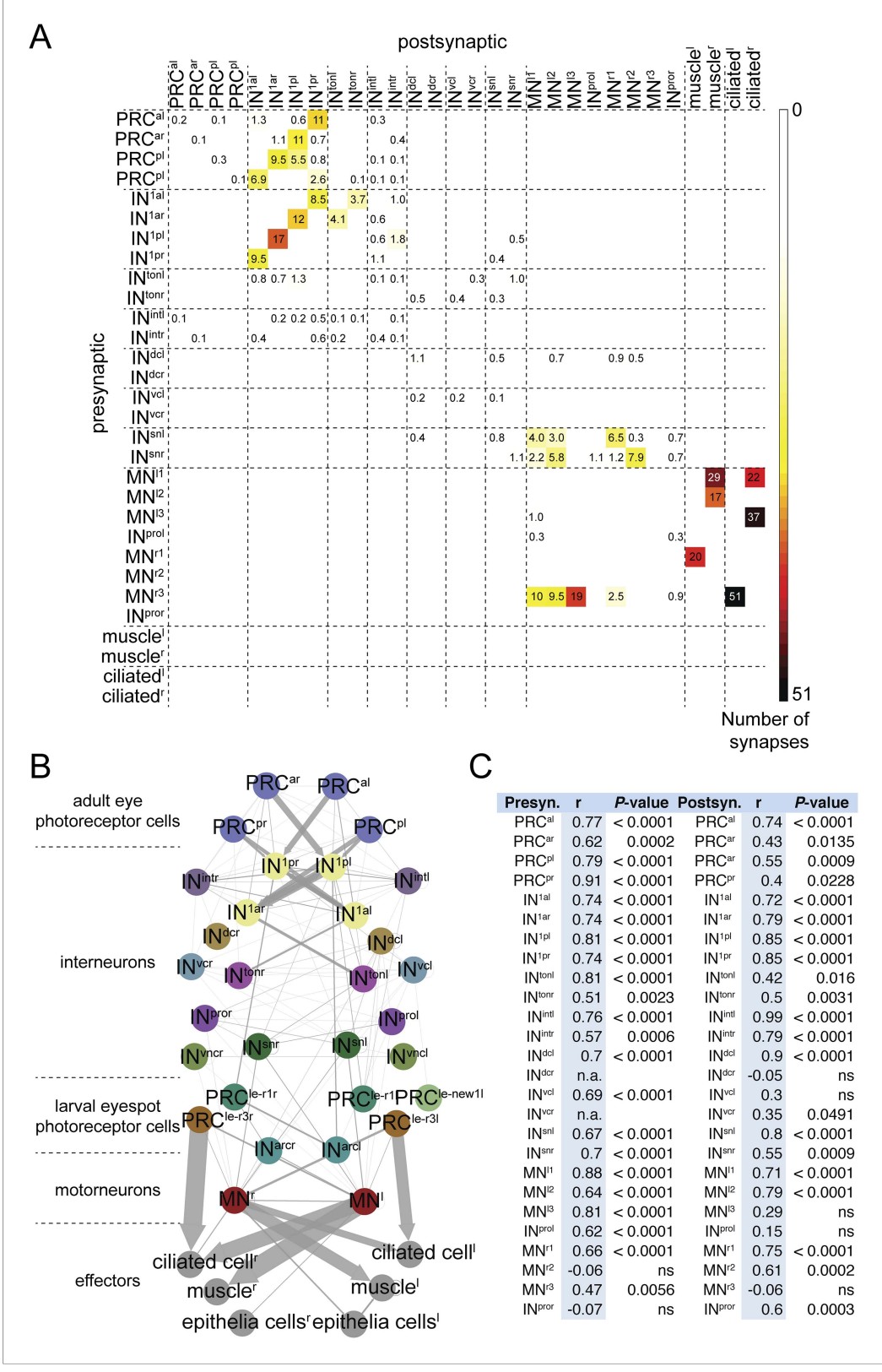

**Figure 2**. Comparative analysis of HT9-3 and HT9-4 visual eye circuits. (**A**) Geometric mean of the neuron types between both individuals, using the averaged synapse number of neuronal groups and the total synapse number on effectors. (**B**) Connectivity graph of the visual and larval eye circuit. Edges are weighted by averaged synapse

*Figure 2. continued on next page*

*Figure 2. Continued*

number (neurons) or by sum of synapses (effectors). Nodes are colored using the color scheme of the neuron types. (**C**) Spearman correlation of the presynaptic and postsynaptic connections of cell groups in HT9-3 and HT9-4. Abbreviations are shown in *Figure 1*.

The following source data and figure supplements are available for figure 2:

**Source data 1**. Full, grouped, and reciprocal connectivity matrices of the eye circuit in HT9-4.

**Figure supplement 1**. Comparison of the neuronal projections and synapse positions of both individuals.

**Figure supplement 2**. Reciprocal connection strength between all neuron pairs in the complete visual circuit (HT9-4).

likely represent incompletely reconstructed $IN^{pro}$ cells.

The new dataset also provides full coverage of all ciliated locomotor cells. The larva has 23 prototroch cells, organized into an anterior and a posterior tier, formed by 11 cell pairs and one unpaired cell (at position 11 o'clock). There are 8 ventral metatroch cells, and three bands of $paratrochs^{I–III}$, one in each segment. The paratrochs have $8^{I}$, $14^{II}$, and $12^{III}$ ciliated cells (*Figure 1B*, *Figure 3E*). We analyzed the innervation of the ciliary bands by MNs in more detail. We found that only the posterior tier of the prototroch receives synapses by MN cells. We also found that several MNs projected to and formed synapses on the paratrochs both on the dorsal and the ventral side (*Figure 3D*, *Source code 1*).

## Larval eyespot connectome reveals sensory-motor circuits of varying complexities

We also fully reconstructed the neuronal circuitry of the larval eyespots or ocelli (structures located ventrally in the head) in HT9-4 (*Figure 4*). The larval eyespots develop in early trochophore larval stages and mediate positive phototaxis in early larval stages by innervating adjacent ciliary band cells (*Jékely et al., 2008*). In trochophore larvae (24–48 hpf), each larval eyespot consists of one pigment cell and one $PRC^{r3}$, the latter of which expresses a rhabdomeric opsin, *r-opsin3*. A preliminary ssTEM analysis in HT9-3 (72 hpf) identified a second $PRC^{r1}$ with an axon projecting towards the brain neuropil (a cerebral PRC), indicating changes in eyespot structure during development. The second PRC expresses another opsin, *r-opsin1* (*Randel et al., 2013*).

Here, we have fully reconstructed the neuronal circuitry downstream of both eyespot PRCs in the 72 hpf larva. The two eyespot PRCs connect to very distinct downstream circuitries, suggesting functional specialization. $PRC^{r3}$ forms *en passant* synapses on ipsilateral ciliary band cells, as previously described for both trochophore and nectochaete stages (*Jékely et al., 2008*; *Randel et al., 2013*). The distal part of the $PRC^{r3}$ axon then turns and meets the secondary optic neuropil, forming synapses on the ipsil- and contralateral MNs. $PRC^{r3}$ is therefore both a direct sensory-MN and a sensory neuron that connects to MNs.

The other eyespot photoreceptor, $PRC^{r1}$, projects into the primary optic neuropil and forms a circuitry distinct from that of $PRC^{r3}$. $PRC^{r1}$ differentiates shortly before 72 hpf, as demonstrated by the late onset of *r-opsin1* expression in this cell (*Randel et al., 2013*). $PRC^{r1}$ is only weakly connected to its target neurons, suggesting that this circuit is not yet fully developed at 72 hpf. However, we identified three cells postsynaptic to $PRC^{r1}$, including the $IN^{1}$ and $IN^{int}$ interneurons of the visual eye circuit and another interneuron, that we named arc interneuron ($IN^{arc}$). $IN^{arc}$ cells synapse on ventral MNs of the visual eye circuitry and two newly identified anterior MNs ($MN^{ant}$), which project to ciliary band cells. We also found 1 or 2 further PRC cells in both eyespots ($PRC^{new}$). These cells had small rhabdoms and short axons with no synapses, indicating that they are still developing.

The connectome of the eyespots suggest that both PRCs share the same effectors with the visual eyes, yet connect to these effectors by distinct circuitry. The eyespots are not able to mediate phototaxis at 72 hpf (*Randel et al., 2014*), but their continued connection to the visual eye circuit suggests that they may play a role in modulating phototaxis or mediating a distinct sensory-motor response.

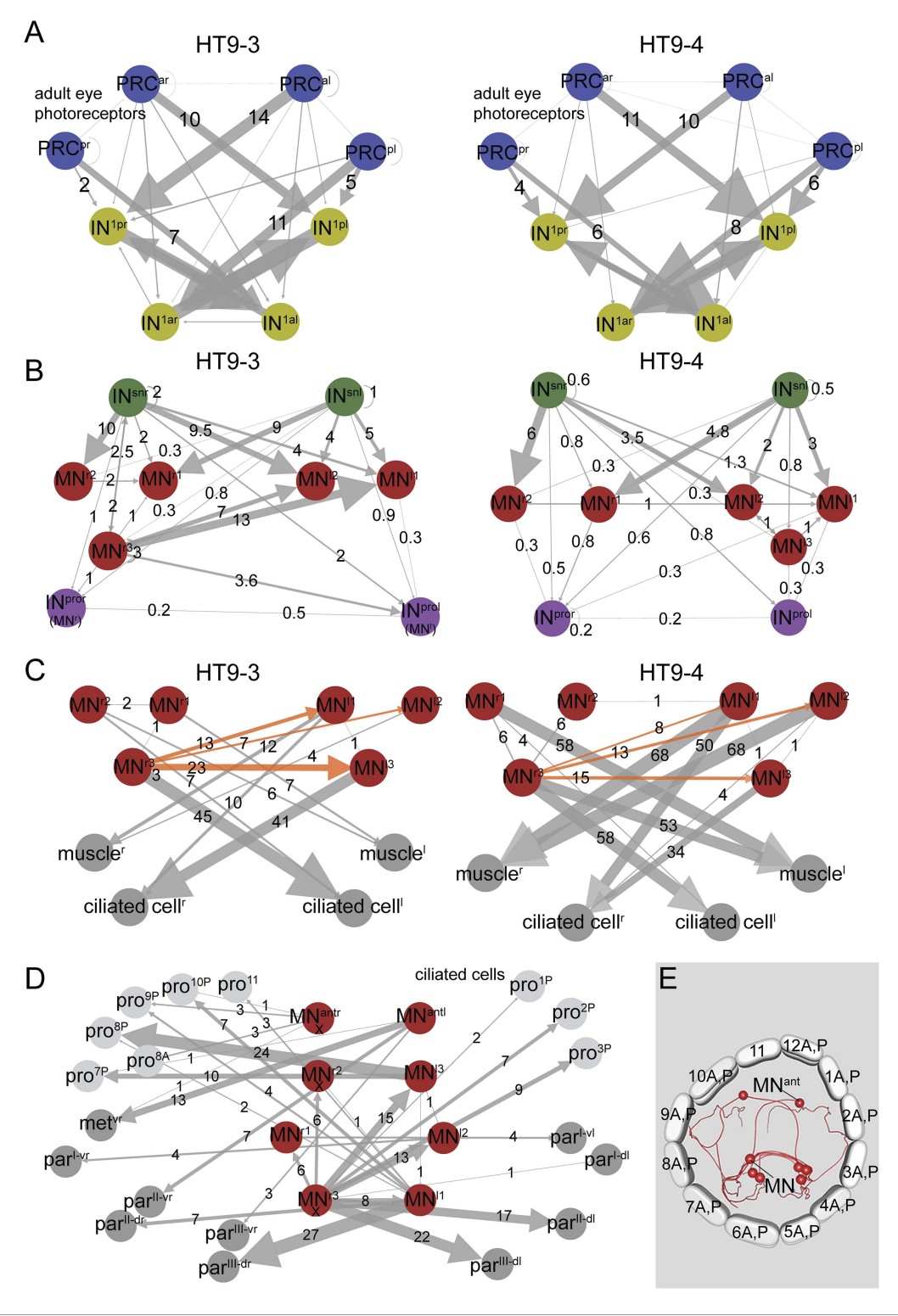

**Figure 3**. Subcircuits of the visual eye connectome of both individuals and the innervation of all ciliary bands in the fully reconstructed second larva (HT9-4). (**A–D**) Connectivity graphs show the connectivity between selected neuron groups and effectors. Edges are weighted by the number of synapses. The number of synapses is also shown for each edge. Nodes are colored using the color scheme of the neuron types. In **A–C**, HT9-3 is left and HT9-4 right. (**A**) Connectivity graph of the grouped PRCs of the visual eyes with IN$^1$ cells. The synapses of the PRCs from each eye are averaged. (**B**) Connectivity graph of IN$^{sn}$ cells with MNs and IN$^{pro}$ cells. The previous nomenclature of the IN$^{pro}$ cells

*Figure 3. continued on next page*

*Figure 3. Continued*

in HT9-3 identified as MNs is shown in brackets. (**C**) Connectivity graph of MNs and their effectors. The synapse numbers shown for ciliated cells and muscles represent the total number of synapses on groups of effector cells. (**D**) Connectivity graph of the MNs of the visual and larval eye circuit of HT9-4 with all ciliary bands. The synapse number of the metatroch and paratrochs are summed. Incompletely traced MNs are labeled with 'X'. (**E**) Blender visualization of the MNs and prototroch cells, apical view. The prototroch cells are numbered in a clockwise fashion. The numbering of the MNs of the left and the right body side within a single larva is not consistent with anatomical pairs, to stick to the nomenclature introduced for HT9-3. Note that the prototroch cell in position 11 is unpaired. met, metatroch; par, paratroch; pro, prototroch; A, anterior; P, posterior. Additional abbreviations are shown in *Figure 1*.

The following figure supplement is available for figure 3:

**Figure supplement 1**. Maturation of photoreceptor connections in HT9-4.

The following source data is available for figure 3s1:

**Figure supplement 1—source data 1**. Development of photoreceptor connectivity in HT9-4.

## Discussion

We have reconstructed the entire larval visual eye circuitry from a second *Platynereis* individual and compared inter-individual connectome stereotypy at the electron microscopic level. We found a high degree of inter-individual stereotypy in the two connectomes at every level of analysis. *Platynereis* larvae develop via a highly stereotypic cell lineage (*Fischer and Arendt, 2013*) and the systematic comparison of gene expression patterns across individuals has revealed the stereotypical position of

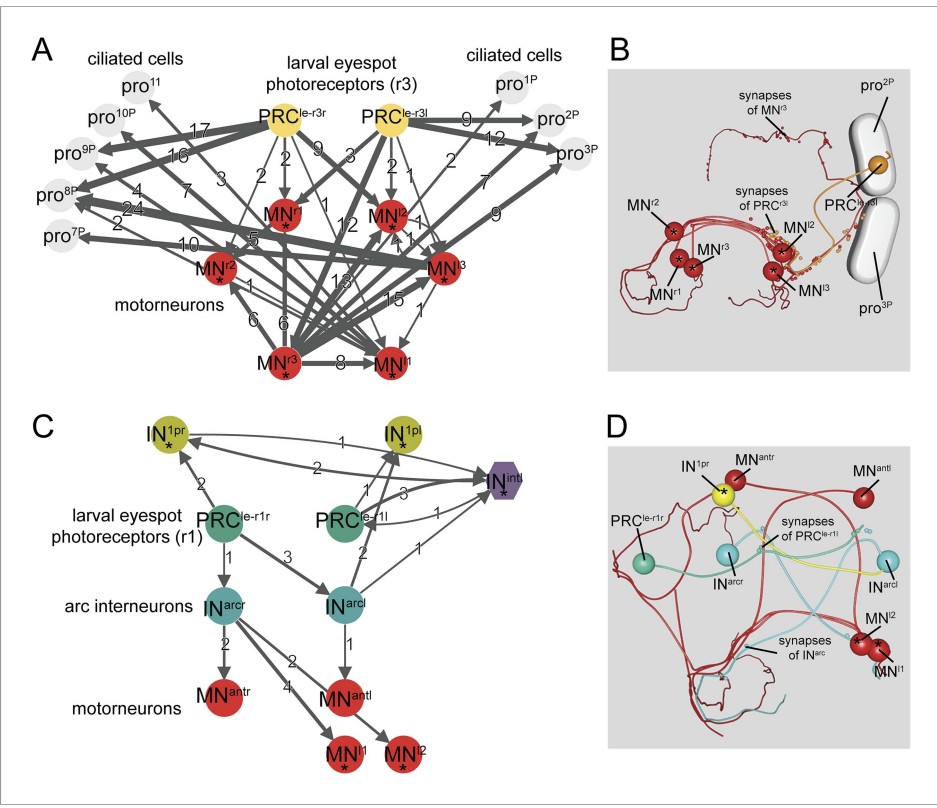

**Figure 4**. Minimal eye circuit of the larval eye. Neurons that are also part of the visual eye circuit are marked by an 'asterisk'. Connectivity graph of the *r-opsin3*-expressing PRC[r3] (**A**) and *r-opsin1*-expressing PRC[r1] (**C**). Connectivity graph edges are weighted by the number of synapses and synapse numbers are shown. Blender visualization of the left PRC[r3l] cell (**B**) and PRC[r1r] circuitry (**D**). Abbreviations are shown in *Figure 1*.

neuronal cell bodies in the 48 hpf trochophore and 72 hpf nectochaete larval stages (*Tomer et al., 2010*; *Asadulina et al., 2012*). Here, we have demonstrated the stereotypy of neuronal projections and synaptic wiring based on ssTEM reconstruction and anatomical comparisons.

The two connectomes are not identical and we could not compare every neuron at a single-cell level. Despite this, clearly identifiable groups of the same cell types could be analyzed. The differences we observed may be due to developmental variation or technical variation. ssTEM datasets are inherently noisy and both datasets contained several neurite fragments that we could not assign to a neuron. However, the fact that our reconstructions resulted in the same overall circuitry indicates that the reconstructions are robust against noise. Our study also demonstrates the benefit of a second dataset when employing an ssTEM approach. Inter-individual comparisons can increase the confidence in weak or asymmetric motifs.

In *Platynereis*, the different sensory-motor circuits of the visual eye PRCs and eyespot PRCs show different levels of complexity. We identified examples for all degrees of sensory-motor contact, from direct sensory-motor innervation through sensory-MN to sensory-IN-MN. The fine gradations in the connectome of the *Platynereis* larval and visual eyes suggest that circuit evolution may have proceeded through the intercalation of new layers of neurons between sensors and effectors.

## Conclusion

Our results indicate that the connectome of the larval visual system in *Platynereis* is highly stereotypical between individuals. These results support our previous findings on stereotypy through the intra-individual analysis of connectivity between neurons on the left and right body sides (*Randel et al., 2014*). Reconstructions from more than one individual can greatly benefit connectome projects.

## Materials and methods

### ssTEM

A 72 hpf *Platynereis* larva (HT9-4) was prepared for serial sectioning and imaging as previously described (*Conzelmann et al., 2013*; *Randel et al., 2014*). Sato's lead citrate was used for contrasting (*Hanaichi et al., 1986*). Images were obtained with SerialEM v3.2.2 (Gatan, Pleasanton, CA) imaging software in conjunction with Digital Micrograph (Gatan). We generated a full-body dataset consisting of 5056 sections, with a thickness of 40 nm per section (*Figure 1—source data 1*). Imaging was performed with a resolution of (5.7 nm/pixel). We lost 1.9% of the sections (98 layers) and the three biggest gaps encompassed five sections. 113 layers distributed across the series were not imaged. We could trace across these gaps using a combination of local and global cues. Additionally, to help the tracing in the neuropil, we re-imaged some of the sections (1229 layers) at high resolution (2.22 nm/pixel) (*Figure 1—source data 1*). Stitching and alignment of the images was carried out using TrakEM2 (*Cardona et al., 2010*). Reconstruction of the cells and connectivity was carried out using Catmaid (*Saalfeld et al., 2009*) (4845 layers). Neurons were traced and reviewed primarily by NR and GJ, with contributions from LABC, CV, and RS.

Catmaid was installed on a Linux container (LXC; linuxcontainers.org) utilizing 8 CPUs (Intel Xeon 2.67 GHz, Intel, Santa Clara, CA) and 8 GB RAM. The server was setup using PostgreSQL (postgresql.org) and nginx/gunicorn (nginx.org; gunicorn.org). The image stacks (1 TB) were served from the same site.

### Neuronal classification, visualization and network analyses

In order to quantify the stereotypy of synaptic connectivity between individuals, we reconstructed the circuitry of the four visual eyes. We traced neurons in synaptic paths downstream of the eye photoreceptors. Cells presynaptic to the eye circuit and weakly connected sensory neurons were not considered. 164 fragments could not be traced (100 fragments < 4 µm, 46 fragments 4–40 µm, and 18 fragments > 40 µm). Fragments are defined as neurons for which a soma could not be found.

We were able to identify the same interneuron and MN types that we reported previously based on the HT9-3 dataset. The stereotypy analysis is mainly based on neuronal groups. For grouped neurons, we used the average synapse number for the calculations. For the effectors, we used the sum of incoming synapses from a group of MNs. Connections were calculated separately for groups of cells with their soma on the left or right side of the body.

Only the primary interneurons and MNs could be compared on a single-cell basis. Anatomical features and neuronal connectivity data were imported into Blender 2.71. Neurons were smoothed such that short branches (<2 μm) were not shown. Network analysis and visualization were performed using Catmaid v0.24, Gephi 0.8.2., prism 5.0, and Blender 2.71.

## Acknowledgements

We thank Elizabeth Williams for comments. We thank Albina Asadulina for help with Blender and Albert Cardona and Tom Kazimiers for help with optimizing Catmaid. The research leading to these results received funding from the European Research Council under the European Union's Seventh Framework Programme (FP7/2007–2013)/European Research Council Grant Agreement 260821.

## Additional information

### Funding

| Funder | Grant reference | Author |
| --- | --- | --- |
| European Research Council (ERC) | Grant Agreement 260821 | Gáspár Jékely |

The funder had no role in study design, data collection and interpretation, or the decision to submit the work for publication.

### Author contributions

NR, GJ, Conception and design, Analysis and interpretation of data, Drafting or revising the article; RS, Acquisition of data, Analysis and interpretation of data, Drafting or revising the article; CV, LAB-C, Analysis and interpretation of data, Drafting or revising the article; SS, Setting up and administering the Catmaid server, Analysis and interpretation of data, Drafting or revising the article

### Author ORCIDs

Gáspár Jékely, http://orcid.org/0000-0001-8496-9836

## Additional files

### Supplementary file

• Source code 1. Blender file containing the visual eye and larval eyespot connectome reconstructed from HT9-4. The file contains the complete anatomical model and connectivity information of the reconstructed visual eye and eyespot circuit from the whole-body dataset of HT9-4. Blender can be downloaded from [http://www.blender.org/]. Display settings can be changed and the connectivity can be analyzed as described in (*Asadulina et al., 2015*).

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
