## [Decision Letter]

Thank you for submitting your work entitled “Inter-individual stereotypy of the *Platynereis* larval visual connectome” for peer review at *eLife*. Your submission has been favorably evaluated by Eve Marder (Senior editor) and two reviewers.

The reviewers have discussed the reviews with one another and the editor has drafted this decision to help you prepare a revised submission.

For your information, we are including the initial reviews of the two reviewers almost in entirety. Please use these to guide your revisions.

*Reviewer #1*:

This manuscript follows up on their previous description of the connectome of the visual sensory-motor circuit in the *Platynereis dumerilii* larva (16). Here the authors address an important issue that was not addressed in their previous work—the extent to which overall anatomical connectivity, as well as synaptic connections at the single cell level, are stereotyped from individual to individual.

The work is done nicely (and it's a lot of work!). Overall, the data suggest a high, but not completely invariant, degree of stereotypical connections between the two examined larvae. This is likely of interest and relevance to researchers involved in doing similar work in other systems, and definitely expands their previous work.

However, I had some difficulty parsing the manuscript. For the general reader interested in knowing how similar (or different) the connectomes of two related individuals are, it would be most useful to rework the manuscript so as to have individual sections highlight the similarities (on a global and local scale), differences (ditto) between the two connectomes, and then lay out the new findings they have made. This would serve to highlight the stereotypy, but also point out the differences clearly. Right now, all this information is buried in a more detailed description of the connectivities and is difficult to extract.

*Reviewer #2*:

This work is part of the heroic effort by the Jekely lab to obtain a complete understanding of the function of the highly simple visual system in a simple marine organism, the larva of *Platynereis*. The same authors have recently published a complete EM reconstruction of the photoreceptors and the network of neurons that process visual information downstream of the different eyes, all the way to the muscles that control movement.

The authors undertook to redo the reconstruction of a similar (second) larva (this time the entire larva although only the visual system is discussed) in order to distinguish between real connectivity and spurious synapses due to mis-alignment or artifacts of the technique. The work, which was presumably initially done as a control, is now used to compare the connectome of two individuals in order to look at stereotypy of connectivity.

The work is of high quality and confirms quite precisely the previous connectome, with different circuits for different photoreceptors. As a byproduct, the authors can now say that the connectome is also highly reproducible, from its general organization all the way to the single cell level. The only differences can be explained by the incomplete reconstruction of the previous larva or to weak connections that were not significant.

The authors conclude that the stereotypy of the visual system is very high. This is not terribly surprising for the cell types/numbers in an animal that follows a very rigorous cell lineage for generating neurons; connectivity is obviously another story but it is accepted that most invertebrates have a highly hard wired nervous system that varies very little with activity or the environment, or even with genetic variations. For instance, there are only few examples of reconfigurations of *Drosophila* neurons in the larva in response to activity. It is true, however that olfactory interneurons and Kenyon cells show extensive random variability but this likely contribute to the sparse odor representation. The recent reconstruction of the *C. elegans* male nervous system does not really address this issue but the data point in this direction.

In conclusion, it is indeed important to check for reproducibility when reconstructing connectomes. The present paper confirms that the initial connectome of the *Platynereis* larva was of very high quality and that the simple network is highly similar in two different organisms. This is nice, but the opposite would have been quite bothersome for the previous paper.

---

## [Author Response]

Reviewer #1:

*[…] Overall, the data suggest a high, but not completely invariant, degree of stereotypical connections between the two examined larvae. This is likely of interest and relevance to researchers involved in doing similar work in other systems, and definitely expands their previous work*.

*However, I had some difficulty parsing the manuscript. For the general reader interested in knowing how similar (or different) the connectomes of two related individuals are, it would be most useful to rework the manuscript so as to have individual sections highlight the similarities (on a global and local scale), differences (ditto) between the two connectomes, and then lay out the new findings they have made. This would serve to highlight the stereotypy, but also point out the differences clearly. Right now, all this information is buried in a more detailed description of the connectivities and is difficult to extract*.

We have reorganized the manuscript. First, we discuss the similarities between the two individuals, then we discuss how the two datasets differ from each other and, finally, we describe the new findings.

Reviewer #2:

*[…] The authors conclude that the stereotypy of the visual system is very high. This is not terribly surprising for the cell types/numbers in an animal that follows a very rigorous cell lineage for generating neurons; connectivity is obviously another story but it is accepted that most invertebrates have a highly hard wired nervous system that varies very little with activity or the environment, or even with genetic variations. For instance, there are only few examples of reconfigurations of* Drosophila *neurons in the larva in response to activity. It is true, however that olfactory interneurons and Kenyon cells show extensive random variability but this likely contribute to the sparse odor representation. The recent reconstruction of the* C. elegans *male nervous system does not really address this issue but the data point in this direction*.

*In conclusion, it is indeed important to check for reproducibility when reconstructing connectomes. The present paper confirms that the initial connectome of the* Platynereis *larva was of very high quality and that the simple network is highly similar in two different organisms. This is nice, but the opposite would have been quite bothersome for the previous paper*.

We thank the reviewer for these comments.